# A new family of cell surface located purine transporters in Microsporidia and related fungal endoparasites

Peter Major[†], Kacper M Sendra, Paul Dean[‡], Tom A Williams[§], Andrew K Watson[#], David T Thwaites, T Martin Embley*, Robert P Hirt*

Institute for Cell and Molecular Biosciences, the Medical School, Newcastle University, Newcastle upon Tyne, United Kingdom

**Abstract** Plasma membrane-located transport proteins are key adaptations for obligate intracellular Microsporidia parasites, because they can use them to steal host metabolites the parasites need to grow and replicate. However, despite their importance, the functions and substrate specificities of most Microsporidia transporters are unknown. Here, we provide functional data for a family of transporters conserved in all microsporidian genomes and also in the genomes of related endoparasites. The universal retention among otherwise highly reduced genomes indicates an important role for these transporters for intracellular parasites. Using *Trachipleistophora hominis*, a Microsporidia isolated from an HIV/AIDS patient, as our experimental model, we show that the proteins are ATP and GTP transporters located on the surface of parasites during their intracellular growth and replication. Our work identifies a new route for the acquisition of essential energy and nucleotides for a major group of intracellular parasites that infect most animal species including humans.

DOI: https://doi.org/10.7554/eLife.47037.001

**\*For correspondence:**
martin.embley@ncl.ac.uk (TME);
robert.hirt@ncl.ac.uk (RPH)

**Present address:** [†]Swiss Tropical and Public Health Institute, Associated Institute of the University of Basel, Basel, Switzerland; [‡]School of Science, Engineering and Design, Teesside University, Tees Valley, United Kingdom; [§]School of Biological Sciences, University of Bristol, Bristol, United Kingdom; [#]Institut de Systématique, Evolution, Biodiversité (ISYEB), Sorbonne Université, CNRS, Museum National d'Histoire Naturelle, EPHE, Université des Antilles, Guadeloupe, France

**Competing interests:** The authors declare that no competing interests exist.

## Introduction

Microsporidia are a highly successful group of strict intracellular eukaryotic parasites that infect a broad range of animal hosts, including humans and economically important species of fish, honeybees, and silkworms (*Stentiford et al., 2016*; *Vávra and Lukeš, 2013*). Microsporidia can only complete their life cycle inside infected eukaryotic cells. In the external environment they exist only as resistant thick-walled spores. New infections are then initiated by spore germination followed by transit of the parasite through a unique polar tube infection apparatus into a eukaryotic host cell (*Vávra and Lukeš, 2013*). As a result of this lifestyle, Microsporidia have undergone dramatic genomic and cellular streamlining including the complete loss of biosynthetic pathways for de novo synthesis of purine and pyrimidine nucleotides (*Dean et al., 2016*; *Dean et al., 2018*; *Heinz et al., 2014*; *Heinz et al., 2012*). At the organelle level, reduction is dramatically illustrated by their minimal mitochondria (called mitosomes) which have lost the organelle genome and the capacity to generate ATP (*Freibert et al., 2017*; *Goldberg et al., 2008*; *Williams et al., 2002*). While glycolysis is conserved in some, but not all (*Wiredu Boakye et al., 2017*) Microsporidia, it appears to be mainly active in spores and is not used during intracellular growth and replication (*Dolgikh et al., 2011*; *Heinz et al., 2012*; *Williams et al., 2014*). The loss of indigenous pathways for making ATP and nucleotides means that Microsporidia are now entirely dependent upon the host cells they infect for the energy, cofactors and nucleic acid building blocks that they need to complete their life cycle (*Dean et al., 2016*; *Nakjang et al., 2013*). Hence parasite transport proteins must play critical roles in servicing the energetic and metabolic demands imposed by parasite growth and replication, but

**eLife digest** Microsporidia are a group of microscopic parasites that spend part of their lives inside the cells of a broad range of animal hosts, including humans. These parasites are considered to be related to fungi, some of which also live within the cells of other species and are known as fungal endoparasites. One of the shared characteristics of these parasites is that they cannot make nucleotides, molecules that are both the main source of energy of the cell and also the building blocks of DNA. Instead, they take nucleotides, or the materials needed to make nucleotides, from their host cells. Once Microsporidia have depleted a host cell, they turn into spores that can survive outside the host until they invade a new cell, starting the cycle anew.

Microsporidia have proteins on their surface, including nucleotide transporter family proteins (NTT), that enable them to import nucleotides from their host into themselves. Although most fungal endoparasites are also thought to steal nucleotides from their hosts, many do not have NTT proteins, raising the question of how they import the nucleotides. A group of proteins called the Major Facilitator Superfamily (MFS) consists of proteins that were thought to transport the materials cells need to make nucleotides (which are also called nucleotide precursors). Members of this family are found throughout Microsporidia and related fungal endoparasites.

These proteins could explain how fungal endoparasites take nucleotides from their hosts. To test this hypothesis, Major et al. infected mammalian cells with Microsporidia and then checked where two MFS proteins were located during infection. This showed that the proteins were on the surface of the endoparasites, implying that they could be nucleotide precursor transporters. Next, Major et al. genetically modified *Escherichia coli* bacteria so they would produce MFS proteins, and showed that the proteins could transport two types of nucleotides. Together these results show that MFS proteins could be responsible for nucleotide transport in fungal endoparasites.

In addition to humans, Microsporidia and related fungal endoparasites infect a wide range of animals, including pollinating insects, which have ecological and economic importance. Given that Microsporidia can only survive if they take nucleotides from their hosts, knowing more about the proteins that import the nucleotides could lead to new cures for Microsporidia infections.
DOI: https://doi.org/10.7554/eLife.47037.002

there is currently little functional data for Microsporidia transporters (*Dean et al., 2016*; *Dean et al., 2014*).

Microsporidia have far fewer (~2000–3500) protein-encoding genes than free-living eukaryotes, providing strong evidence that gene loss, particularly for metabolism, is a pervasive feature of their evolution (*Cuomo et al., 2012*; *Nakjang et al., 2013*). Genes for candidate surface-located transporters that have been retained against this background of gene loss, are likely to be important for supporting parasite growth and replication (*Heinz et al., 2012*; *Nakjang et al., 2013*). We previously identified 10 gene families encoding candidate surface transport proteins that are conserved in all published Microsporidia genomes (*Heinz et al., 2012*; *Nakjang et al., 2013*). The conserved families include the experimentally characterised nucleotide transporters (NTT) that are expressed on the cell surface of intracellular Microsporidia where they can import host purine nucleotides, including ATP, GTP and NAD$^+$ (*Dean et al., 2016*; *Dean et al., 2014*; *Dean et al., 2018*; *Tsaousis et al., 2008*). Intriguingly, the NTT transporters did not transport pyrimidine nucleotides in these experiments (*Dean et al., 2018*; *Heinz et al., 2014*), even though genome analyses suggest that Microsporidia can no longer make these substrates for themselves (*Dean et al., 2016*; *Dean et al., 2018*; *Heinz et al., 2014*; *Tsaousis et al., 2008*). These data suggest that additional transporters must exist to supply Microsporidia with the pyrimidines that they need to grow and replicate.

In the present study, we have characterised a family of Microsporidia Major Facilitator Superfamily (MFS) transport proteins which were discovered in *Nematocida* spp. (*Cuomo et al., 2012*) and subsequently shown to be present in all Microsporidia for which genomes are available (*Cuomo et al., 2012*; *Dean et al., 2018*; *Heinz et al., 2014*; *Heinz et al., 2012*; *Nakjang et al., 2013*; *Watson et al., 2015*). Some of the *Nematocida* proteins share a Pfam domain (PF03825) with the NupG transporter of *Escherichia coli* (*Xie et al., 2004*), suggesting (*Cuomo et al., 2012*) that, like NupG, they might be purine and pyrimidine nucleoside transporters, potentially solving the parasite

pyrimidine deficit discussed above. To test this hypothesis and to investigate the evolution and role (s) of these MFS transporters in Microsporidia, we characterised the expression, cellular location and functional characteristics of the homologous proteins from *Trachipleistophora hominis* (*Heinz et al., 2012*; *Nakjang et al., 2013*; *Watson et al., 2015*), a model species that can be maintained in cell culture and was originally isolated from an HIV/AIDS patient (*Field et al., 1996*). Our data provide no evidence that the *T. hominis* proteins can transport the pyrimidine nucleoside uridine, a known substrate for NupG (*Xie et al., 2004*), or the pyrimidine nucleotides CTP or UTP, but demonstrate that they do transport the purine nucleotides ATP and GTP. These data reveal that *T. hominis*, and potentially other Microsporidia, have at least two distinct transport systems for importing the ATP and GTP from infected host cells, that they need to complete their intracellular lifecycles.

## Results and discussion

### Microsporidia encode multiple paralogues of MFS transporters that are distantly related to *E. coli* NupG

We used BlastP to search for sequences related to the Microsporidia conserved protein family # c_456 (*Nakjang et al., 2013*) which contains the *Nematocida* spp. putative NupG-like nucleoside/ H+ symporter (*Cuomo et al., 2012*), in the genomes of Microsporidia and their endoparasitic relatives among the Rozellomycota (*Corsaro et al., 2016*), including species of *Rozella* (*James et al., 2013*), *Mitosporidium* (*Haag et al., 2014*) and *Amphiamblys*, (*Mikhailov et al., 2017*). Recent phylogenetic analyses suggest that Rozellomycota is a paraphyletic group containing a 'core Microsporidia' clade (*Corsaro et al., 2016*), although others have argued for a different taxonomy with an expanded definition for the Microsporidia (*Bass et al., 2018*). The core Microsporidia (the term we use here) contains all the species traditionally classified as Microsporidia and which share diagnostic features of the group, including a coiled polar tube, a polaroplast, and the absence of a mitochondrial genome (*Corsaro et al., 2016*; *Mikhailov et al., 2017*). A total of 63 members of the protein family c_456 were identified among the Rozellomycota and core Microsporidia (*Figure 1—source data 1*). Phylogenetic analysis including homologues from prokaryotes and eukaryotes recovered the Microsporidia transporters as a separate clade with homologues from *Mitosporidum*, *Amphiamblys* and *Rozella* (*Figure 1*, *Figure 1—figure supplements 1* and *2*); providing evidence that the genes were present in the common ancestor of these species. Homologues of the Microsporidia proteins were also detected in other eukaryotes including free-living fungi, Oomycetes, Metazoa, Euglenozoa, Alveolata and Stramenopiles. The broad distribution of the genes across major lineages of eukaryote including free-living species suggests that this family of transport proteins may be ancestral among eukaryotes. In particular, the tree topology provides no compelling support for an origin of the Microsporidia genes through LGT from prokaryotes as previously suggested (*Cuomo et al., 2012*), and there is no evidence from the tree for a specific relationship to the *E. coli* NupG and XapB (*Nakjang et al., 2013*; *Xie et al., 2004*) transporters (*Figure 1—figure supplements 1* and *2*).

Detailed phylogenetic analysis identified two clades (A and B) of Microsporidia proteins (*Nakjang et al., 2013*) originating from a gene duplication in the common ancestor of the core Microsporidia and some Rozellomycota (*Figure 1*, *Figure 1—figure supplement 1*). Clade A contains at least one copy from every Microsporidia genome sampled. Clade B contains sequences from most Microsporidia apart from the *Encephalitozoon/Nosema/Ordospora* and *Enterocytozoon/Vittaforma* lineages, which appear to have lost the genes for these transporters (*Figure 1*). The majority of the Microsporidia sequences from clade A contain an indel (insertion or deletion) between the 7th and 8th transmembrane domains (TMD) compared to other proteins (*Nakjang et al., 2013*) (*Figure 1—figure supplement 3*). The indel is between 16 to 38 residues long (*Figure 1—figure supplement 3*) apart from one of the paralogues from *Anncaliia algerae* which has lost most of the indel (*Figure 1—figure supplement 3*). The sequences from *Rozella*, *Mitosporidium* and *Amphiamblys* at the base of clade A do not possess the indel (*Figure 1—figure supplement 3*). Clades A and B both contain evidence for further lineage-specific duplications affecting some Microsporidia (*Figure 1*). For example, in clade A, *T. hominis* and some other microsporidian genomes encode only a single gene, whereas others have multiple paralogues (*Figure 1*, *Figure 1—source data 1*). In clade B, *T. hominis* has three paralogues, whereas other species have only a single gene or appear to have lost their clade B homologue altogether (*Figure 1*). This type of lineage-specific gene duplication is a

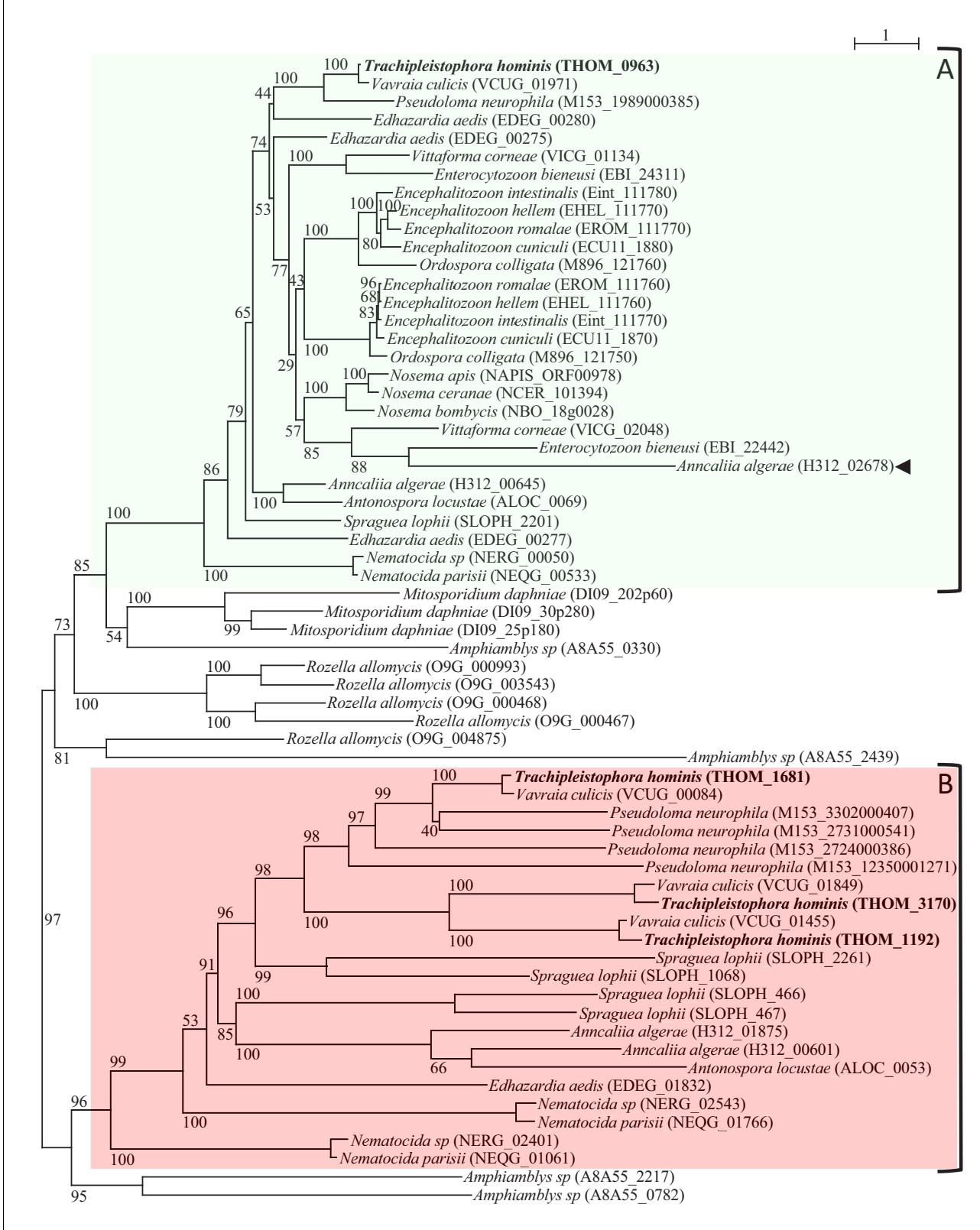

**Figure 1.** Protein maximum-Likelihood phylogeny of MFS transporters from Microsporidia and close relatives.  The MFS sequences from the core Microsporidia (see *Corsaro et al., 2016*; *Galindo et al., 2018*) and their close relatives from the genus *Amphiamblys*, *Mitosporidium*, and *Rozella* clustered in two distinct clades based on the rooting inferred from a broader analysis (*Figure 1—figure supplements 1* and *2*). Microsporidia genomes encode at least one member of clade A (highlighted with the green box). All Microsporidia sequences (except one - arrowhead) in clade A possess a

*Figure 1 continued on next page*

*Figure 1 continued*

distinctive indel (see main text and *Figure 1—figure supplement 3*). Some sequences from the Rozellomycota including *Rozella allomycis*, *Mitosporidium daphniae* and *Amphiamblys* sp. also cluster with Microsporidia clade A or clade B sequences (the Microsporidia clade B is highlighted with the red box). A number of Microsporidia and Rozellomycota species appear to have lost their clade B homologues. Lineage-specific expansions can be observed for several species in clade A or B. The maximum likelihood phylogeny was inferred with the LG+C60 model in IQ-TREE with ultrafast bootstrap branch support values (1000 replicates). The scale bar (top right) represents the number of inferred amino acid changes per site.

DOI: https://doi.org/10.7554/eLife.47037.003

The following source data and figure supplements are available for figure 1:

**Source data 1.** MFS protein sequences from Microsporidia and Rozellomycota analysed in this study.
DOI: https://doi.org/10.7554/eLife.47037.008

**Source data 2.** MFS homologues from the family c_456 (*Nakjang et al., 2013*) encoded by the genome of two recently sequenced Rozellomycota species.
DOI: https://doi.org/10.7554/eLife.47037.009

**Figure supplement 1.** Maximum likelihood phylogeny of homologues to ThMFS proteins including a broad range of eukaryotes and prokaryotes.
DOI: https://doi.org/10.7554/eLife.47037.004

**Figure supplement 2.** Detailed phylogeny of MFS proteins shown in *Figure 1—figure supplement 1*.
DOI: https://doi.org/10.7554/eLife.47037.005

**Figure supplement 3.** Comparisons of indels in the extracellular loop between the 7th and 8th transmembrane domains present in Microsporidia MFS proteins from clade A.
DOI: https://doi.org/10.7554/eLife.47037.006

**Figure supplement 4.** Predicted topology of the ThMFS1-4 proteins.
DOI: https://doi.org/10.7554/eLife.47037.007

feature of the evolution of other Microsporidia transporters (*Nakjang et al., 2013*) providing the raw material (new genes) for functional divergence among paralogues (*Dean et al., 2018*; *Heinz et al., 2014*).

We used the four *T. hominis* sequences and related proteins from Microsporidia and Rozellomycota as search queries in HMMer searches against the Pfam domain database (*Figure 1—source data 1*). The Microsporidia members of clade A, one sequence from *Mitosporidium daphniae* and four from *Rozella allomycis* matched the MFS profile PF03825 found in nucleoside/H$^+$ symporters including *E. coli* NupG. This profile did not match any of the other transporter homologues, including the proteins from *Amphiamblys* sp. (*Figure 1—source data 1*). Microsporidia sequences from clade B were characterised by matches to the more general MFS transporter profiles MFS_1 (PF07690) and MFS_2 (PF13347) (*Figure 1—source data 1*). The alignment for the indel in sequences from clade A did not generate a significant hit (all e-values $\geq$ 13) in a HHPred profile-profile search of gene families in the protein domain databases (data not shown). The Microsporidia MFS transporters are predicted to have 12 alpha helical TMD, with both termini facing the cytoplasmic side (*Figure 1—figure supplement 4*). This is a typical conformation (*Reddy et al., 2012*) as previously reported for the *E. coli* NupG and XapB nucleoside/H+ symporters (*Nørholm and Dandanell, 2001*; *Vaziri et al., 2013*; *Xie et al., 2004*) and is consistent with the Microsporidia proteins being functional MFS transporters.

In summary, all the Microsporidia proteins are members of the major facilitator family (MFS) of transport proteins, but there is no compelling evidence for a particularly close relationship to NupG, or strong indication from trees or bioinformatics concerning their substrates or transport mechanisms. In the rest of the manuscript we refer to the four *T. hominis* transporters as *T. hominis* ThMFS1-4 with the following locus tags and uniprot accessions (*Heinz et al., 2012*): in clade A, ThMFS1: THOM_0963 - L7J $\times$ 55; in clade B, ThMFS2: THOM_1192 - L7J $\times$ 19; ThMFS3: THOM_1681 - L7JVD0; ThMFS4: THOM_3170 - L7JT12.

## ThMFS transporters are expressed during infection and localize to the cell surface of intracellular parasites

We used RNA-Seq and measured transcript abundance (expressed as transcripts per million, TPM) at six time points representing different stages in the parasite lifecycle, during a synchronised intracellular infection of rabbit kidney (RK13) cells by *T. hominis* (*Dean et al., 2018*). ThMFS1 mRNA was characterised by a relatively high abundance at the first time point (3 hr) followed by a gradual decrease at 14, 22 and 40 hr (*Figure 2A*) and then a dramatic increase after 40 hr during spore

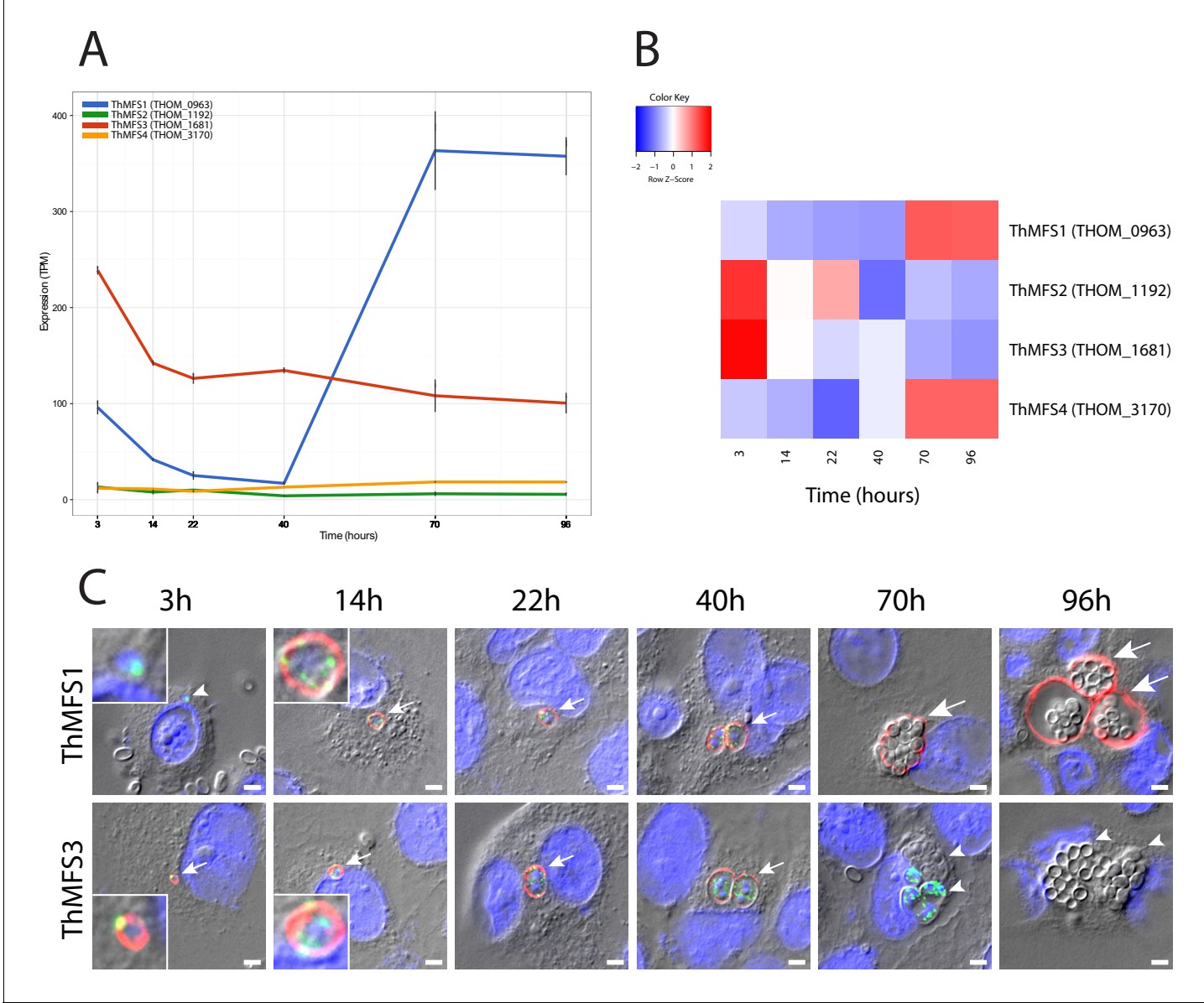

**Figure 2.** Transcription and subcellular localization of *T. hominis* MFS- proteins in a synchronised infection. (**A**) RNA-Seq data for the four ThMFS1-4 encoding genes for six time points post infection corresponding to key stages of the *T. hominis* infection cycle. The Y-axis shows transcripts per million reads (TPM - Black lines indicate the value from each replicate) against time on the X-axis (hours). (**B**) Relative abundance of transcripts (Z-score - normalised values based on the average TPM for each gene) for each ThMFS gene across the time points illustrated in panel A. (**C**) IFA data for ThMFS1 (THOM_0963) (rabbit 94, antisera dilution 1:50, red) and ThMFS3 (THOM_1681) (rabbit 91, antisera dilution 1:50, red) proteins showing localization to the periphery of parasites. Time points were chosen based on the following stage-specific features appearing post infection: 3 hr germinated sporoplasm (smallest vegetative cells of the parasite, see inset for zoom in on the parasite), 14 hr unicellular meronts (see inset for zoom in on the parasite), 22 hr first nuclear division, 40 hr first cellular division, 70 hr initiation of cellular differentiation into sporonts and spores, 96 hr fully mature spores within the SPOV (*Dean et al., 2018*). Small arrows indicate labelled parasites, large arrows indicate labelled SPOV, small arrow heads indicate unlabelled parasites (3 hr and 70 hr) or unlabelled SPOV (96 hr). Infection of new host cells from mature spores can be observed in the later time points (an example is illustrated in *Figure 2—figure supplement 9*). ThMFS1 was not detectable at the first time point whereas the sporoplasms were labelled with the ThmitHsp70 mitosomal marker (rat antisera dilution 1:200, green). Quantification of the different IFA signals (ThMFS1, ThMFS3 and mitHsp70) (*Figure 2—source data 3*, *Figure 2—figure supplement 2*) is consistent with the pattern observed in panel 2C. The nuclei of the mammalian host cells (large nuclei) and parasites (small nuclei) were labelled with DAPI (blue). The scale bar is 2 μm.

DOI: https://doi.org/10.7554/eLife.47037.010

The following source data and figure supplements are available for figure 2:

**Source data 1.** SNP analysis of the ThMFS1-4 ORFs.

*Figure 2 continued on next page*

*Figure 2 continued*

DOI: https://doi.org/10.7554/eLife.47037.020

**Source data 2.** Transcriptomics data for ThMFS1-4 and ThNTT1-4 transporters.

DOI: https://doi.org/10.7554/eLife.47037.021

**Source data 3.** Quantifications of IFA signals for ThMFS1-4 and mitHsp70.

DOI: https://doi.org/10.7554/eLife.47037.022

**Figure supplement 1.** Published RNA-Seq data from the Microsporidia *Vavraia culicis*, *Encephalitozoon cuniculi*, *Nematocida parisii* and *Edhazardia aedis*.

DOI: https://doi.org/10.7554/eLife.47037.011

**Figure supplement 2.** Comparison of IFA signal quantification for ThMFS1, ThMFS3 and mitHsp70.

DOI: https://doi.org/10.7554/eLife.47037.012

**Figure supplement 3.** Peptide designed to generate specific antisera for the ThMFS1-4 proteins.

DOI: https://doi.org/10.7554/eLife.47037.013

**Figure supplement 4.** Comparison of the IFA signals for antisera for ThMFS1-4 and mitHsp70.

DOI: https://doi.org/10.7554/eLife.47037.014

**Figure supplement 5.** Western blot analysis on total protein extracts from host cells and parasites with antisera against ThMFS1-4.

DOI: https://doi.org/10.7554/eLife.47037.015

**Figure supplement 6.** Dot blots on peptides to test the specificity of anti-ThMFS1-4 rabbit antibodies.

DOI: https://doi.org/10.7554/eLife.47037.016

**Figure supplement 7.** ThMFS3 IFA detection in highly infected RK13 cells containing mixed stages of the parasite infection cycle.

DOI: https://doi.org/10.7554/eLife.47037.017

**Figure supplement 8.** Evidence for re-infections from germination of newly formed mature spores in the late time point post infection (96 hr) from IFA for ThMFS1 and ThMFS3.

DOI: https://doi.org/10.7554/eLife.47037.018

**Figure supplement 9.** Peptide competition experiments demonstrate the specificity of antisera against the ThMFS1 and ThMFS3 proteins.

DOI: https://doi.org/10.7554/eLife.47037.019

formation (70 hr) and maturation (96 hr), when it was the most abundant ThMFS mRNA (peaking with a mean value of 363 TPM, *Figure 2A*). A previous RNA-Seq analysis of a sample taken at a late time point in a *T. hominis* infection of RK13 cells, also identified ThMFS1 as the most abundant transcript among the four ThMFS genes (*Watson et al., 2015*). By contrast, the expression profile of ThMFS3 mRNA peaked during the early stages of infection at the 3 hr post-infection time-point (238 TPM), and gradually declined to about 50% of this level by the 22 hr time point (*Figure 2A*). ThMFS2 and ThMFS4 were both characterised by significantly lower levels of transcript abundance throughout the infection, with mean levels at all time points below 19 TPM. Relative expression profiles for ThMFS1-4 (*Figure 2B*) further emphasise the contrasting levels of transcripts for the different transporters during infection. Published RNA-Seq data for other Microsporidia (*Cuomo et al., 2012*; *Desjardins et al., 2015*; *Grisdale et al., 2013*) show similar patterns in the expression of paralogous MFS-like genes (*Figure 2—figure supplement 1*). For example, the *Vavraia culicis* orthologue of ThMFS1 (*Figure 1*) shows its highest expression during spore formation (*Figure 2—figure supplement 1*) and the *V. culicis* orthologues for ThMFS2 and ThMFS4 (*Figure 1*) have the lowest levels of transcription (*Figure 2—figure supplement 1*). Based upon the branch lengths in the phylogenetic tree (*Figure 1*), ThMFS2 and ThMFS4 are evolving much faster than ThMFS3 (*Figure 1*) and ThMFS2 and ThMFS4 also have the highest number of single-nucleotide polymorphisms (SNPs) among the ThMFS homologues (*Figure 2—source data 1*). Previous work (*Dean et al., 2018*) on the *T. hominis* NTT (ThNTT) demonstrated a similar reciprocal relationship between the degree of ThNTT sequence conservation (expressed as tree branch lengths) and expression levels, with the fastest evolving ThNTT paralogues showing the lowest expression levels. Comparing ThMFS and ThNTT (*Dean et al., 2018*) transcript abundance indicated that ThMFS1 and ThMFS3 are characterised by higher levels of expression than ThNTT1-3 throughout the time course of the experiments (*Figure 2—source data 2*). Notably, ThNTT4 is characterised by the highest level of transcripts among all ThMFS and ThNTT transporters, with mean TPM values of >1000 across six different time points of the synchronised infection (*Figure 2—source data 2*) (*Dean et al., 2018*).

We made anti-peptide (*Figure 2—figure supplement 3*) rabbit antibodies for each of the four *T. hominis* MFS transporters and used them in immunofluorescence assays (IFA) of infected cells grown on slides and sampled at different time points (*Figure 2C*). Each slide was co-incubated with rabbit

antisera from one of the ThMFS (red) and rat antisera to *T. hominis* mitochondrial Hsp70 (ThmitHsp70) (green), with the latter labelling parasite mitosomes and acting as a control for the fixation and permeabilization protocol (*Freibert et al., 2017*; *Goldberg et al., 2008*). DAPI (4',6-diamidino-2-phenylindole) (blue) was also added to stain parasite and host nuclear DNA (*Figure 2C*). During intracellular infection (*Field et al., 1996*) the surface plasma membrane of *T. hominis* is exposed to the host cell cytosol or to the lumen of an intracellular compartment called a sporophorous vesicle (SPOV) (*Cali and Tokvarian, 2014*). In the infective spore stage, the plasma membrane is protected from the external environment by a thick spore coat (*Vávra and Lukeš, 2013*). The antisera to ThMFS1 and ThMFS3 gave clear and specific labelling of the surface of parasites consistent with a plasma membrane location (*Heinz et al., 2014*) in samples taken during the time course of infection (*Figure 2C*). By contrast, no parasite specific IFA signals were detected for the antisera to ThMFS2 and ThMFS4 at any stage of the infection cycle, despite observing consistent labelling of the mitosomes and nuclei of parasites (*Figure 2—figure supplement 4*). The reasons for the lack of ThMFS2 and ThMFS4 signals are not known but the expression levels of these two genes were much lower than for ThMFS1 and ThMFS3 (*Figure 2A*), so it is possible that the antibodies we used were insufficiently sensitive to detect expression in our model system or the proteins were not expressed under the experimental conditions used. Consistent with these possibilities, the antisera for ThMFS2 and ThMFS4 gave no parasite-specific signal in western blot analyses on total protein extracts from *T. hominis* infected RK13 cells and *T. hominis* spores, in contrast to the strong signals for antisera to ThMFS1 and ThMFS3 (*Figure 2—figure supplement 5*). The antisera for ThMFS2 and ThMFS4 also generated relatively weaker signals against the peptides used as antigens in dot blots suggesting the antibodies may be ineffective (*Figure 2—figure supplement 6*).

The antisera for ThMFS1 and ThMFS3 showed different patterns of parasite labelling during the infection (*Figure 2C*), suggesting that they have evolved functional differences following gene duplication, as previously shown for the paralogous NTT transporters of *T. hominis* (*Dean et al., 2018*). We detected no labelling of the parasite surface by the antisera to ThMFS1 in replicate slides at 3 hr post-infection, despite transcript levels for the gene being relatively high at this point (*Figure 2A*) and clear labelling of mitosomes and nuclei on the same slides (*Figure 2C*). The ThMFS1 antisera gave strong labelling of the surface of parasites at 14 hr, 22 hr and 40 hr post-infection (*Figure 2C*). At 70 hr and 96 hr post-infection, strong signal was located in the SPOV membrane surrounding groups of *T. hominis* (*Figure 2C*) sporonts and spores (*Hollister et al., 1996*; *Watson et al., 2015*). This suggests that ThMFS1 has a role in supporting parasite development within the SPOV. The spores inside the SPOV were not labelled with either of the antibodies or by DAPI, suggesting that the developing spore coat may exclude access of these reagents. A lack of signal from spores in these types of experiments has previously been observed using antibodies to the ThNTT transporters and to several mitosomal proteins (*Dean et al., 2018*; *Freibert et al., 2017*; *Goldberg et al., 2008*). Proteomics data for purified *T. hominis* spores (*Heinz et al., 2012*) has shown, however, that ThMFS1 is present in spores. We detected strong parasite surface labelling by antisera to ThMFS3 from 3 hr up to 40 hr post-infection, but no labelling by antisera to ThMFS3 of either parasites or SPOV membrane in slides for 70 hr and 96 hr (*Figure 2C*). The absence of signal for ThMFS3 on the surface of mature vegetative cells at 70 hr appears to be stage specific because fields of highly infected RK13 cells that contain a mix of mature and early stage parasites (through reinfection), show strongly labelled smaller cells (*Figure 2—figure supplement 7*). Labelling of small cells characteristic of earlier time points, with antisera for ThMFS1 and ThMFS3 at 96 hr post-infection, are also consistent with the initiation of new infections from germinating newly differentiated spores (*Figure 2—figure supplement 8*). Consistent with the absence in IFA of a detectable signal for ThMFS3 in the later stages of parasite development, ThMFS3 was not detected in proteomics data for purified *T. hominis* spores (*Heinz et al., 2012*).

## *Trachipleistophora hominis* MFS proteins transport purine nucleotides but not uridine

Upon their discovery, it was suggested (*Cuomo et al., 2012*) that the *Nematocida* homologues of ThMFS transporters might be purine and pyrimidine nucleoside transporters like the *E. coli* NupG transporter. NupG has a broad specificity for nucleosides (*Nørholm and Dandanell, 2001*; *Patching et al., 2005*; *Vaziri et al., 2013*; *Xie et al., 2004*) including the pyrimidine nucleoside uridine (*Nørholm and Dandanell, 2001*; *Xie et al., 2004*). We used heterologous expression in *E. coli*

strains (*Dean et al., 2018*; *Heinz et al., 2014*; *Tsaousis et al., 2008*) to test if the ThMFS1-4 proteins could transport radiolabelled uridine, using the *E. coli* NupG transporter as a positive control. Because expression of eukaryotic proteins in bacteria can be improved by codon optimisation in the bacterial host (*Gustafsson et al., 2012*) we tested expression of native and *E. coli* codon-optimised ThMFS in different *E. coli* strains. The *E. coli* strain GD1333, which lacks the two endogenous nucleoside transporters NupG and NupC (*Nørholm and Dandanell, 2001*), was used as the expression host in these assays (*Figure 3A*). In contrast to the positive control expressing recombinant NupG, none of the ThMFS proteins transported [$^{14}$C]-uridine above background levels for the empty vector control (ptrc99a) (*Figure 3A*). These data suggest that uridine is not a substrate for transport by ThMFS1-4.

We recently characterised the substrate specificities and evolution of a family of Microsporidia nucleotide transport proteins (NTT, Pfam profile PF03219) that import ATP and GTP for parasite growth and replication inside infected host cells (*Dean et al., 2018*; *Heinz et al., 2014*). These transporters (family # c_336 in *Nakjang et al., 2013*) are also members of the MFS protein superfamily (clan CL0015; *Dean et al., 2018*). To investigate if the structural similarity between the ThMFS and NTT transporters is reflected in their transport properties, we tested if the ThMFS proteins could transport radiolabelled purine and pyrimidine nucleotides when expressed in *E. coli* (*Figure 3B and C*). All four ThMFS transported ATP above background, whereas *E. coli* expressing NupG did not transport ATP (*Figure 3B*). Further experiments demonstrated that all four ThMFS proteins can also transport radiolabelled-GTP, but not radiolabelled-CTP or UTP (*Figure 3C*). Consistent with the specific uptake of [$\alpha^{32}$P]-ATP and [$\alpha^{32}$P]-GTP, uptake of both substrates by *E. coli* cells expressing ThMFS transporters was time-dependent (*Figure 4*). Our data demonstrate that ThMFS1-4 can transport purine and thus their substrate specificity overlaps with that of the previously characterised Microsporidia NTT nucleotide transporters (*Dean et al., 2018*; *Heinz et al., 2014*). The lack of a positive control in these assays makes it difficult to interpret the negative results for transport of the two tested pyrimidines.

The *E. coli* NupG transporter is a H$^+$ symporter that uses a proton gradient to drive the transport of uridine and other nucleosides (*Xie et al., 2004*). We have previously shown that some Microsporidia NTTs have evolved into symporters (*Dean et al., 2018*) capable of mediating the net import of nucleotides. To test if nucleotide uptake by ThMFS proteins is also proton dependent, we investigated [$\alpha^{32}$P]-ATP and [$\alpha^{32}$P]-GTP uptake by ThMFS1 and ThMFS3 in the presence of the protonophore carbonyl cyanide m-chlorophenyl hydrazone (CCCP). ThMFS1 and ThMFS3 are the most abundant transcripts in RNA-Seq (*Figure 2A*) and both are detected on the cell surface of actively growing parasites (*Figure 2C*). The addition of CCCP had no effect on nucleotide uptake by *E. coli* expressing ThMFS1 or ThMFS3 (*Figure 5*), suggesting that they are not H$^+$-symporters. By contrast, CCCP did inhibit import of nucleotides by *Protochlamydia amoebophila Pam*NTT5, a known H$^+$-symporter of GTP (*Haferkamp et al., 2006*) (*Figure 5*). The nucleotide transport activity of some Microsporidia NTT (e.g. ThNTT4) were previously shown to be sensitive to CCCP treatment in the same expression system (*Dean et al., 2018*).

## Conclusions

Previous studies have shown that Microsporidia lack the genes needed to make primary metabolites including nucleotides, and that they have a limited capacity to make their own energy (*Dean et al., 2016*; *Williams et al., 2014*). This raises the question of how these intracellular parasites obtain the enormous amounts of ATP and other nucleotides that they need to support their rapid growth and replication. For example, it is suggested that it takes at least 10$^9$ ATPs (*Phillips and Milo, 2009*) to make a single *E. coli*, and the ATP requirement to make the larger cells of Microsporidia is likely to be considerably higher. Previous work suggests that Microsporidia can use surface-located NTT nucleotide transporters to import purine nucleotides including ATP and GTP during their intracellular growth (*Dean et al., 2016*; *Dean et al., 2018*; *Tsaousis et al., 2008*). In the present study, we show that *Microsporidia* have a second set of MFS transporters that they can potentially use to supplement their energy and nucleotide budget. Thus, all four *T. hominis* ThMFS transporters were able to transport ATP and GTP when expressed in *E. coli* (*Figure 3C*), and the two most highly expressed proteins (ThMFS1 and ThMFS3) are present on the surface of parasites (along with the NTTs; *Dean et al., 2018*) when they are most actively growing inside infected rabbit kidney cells

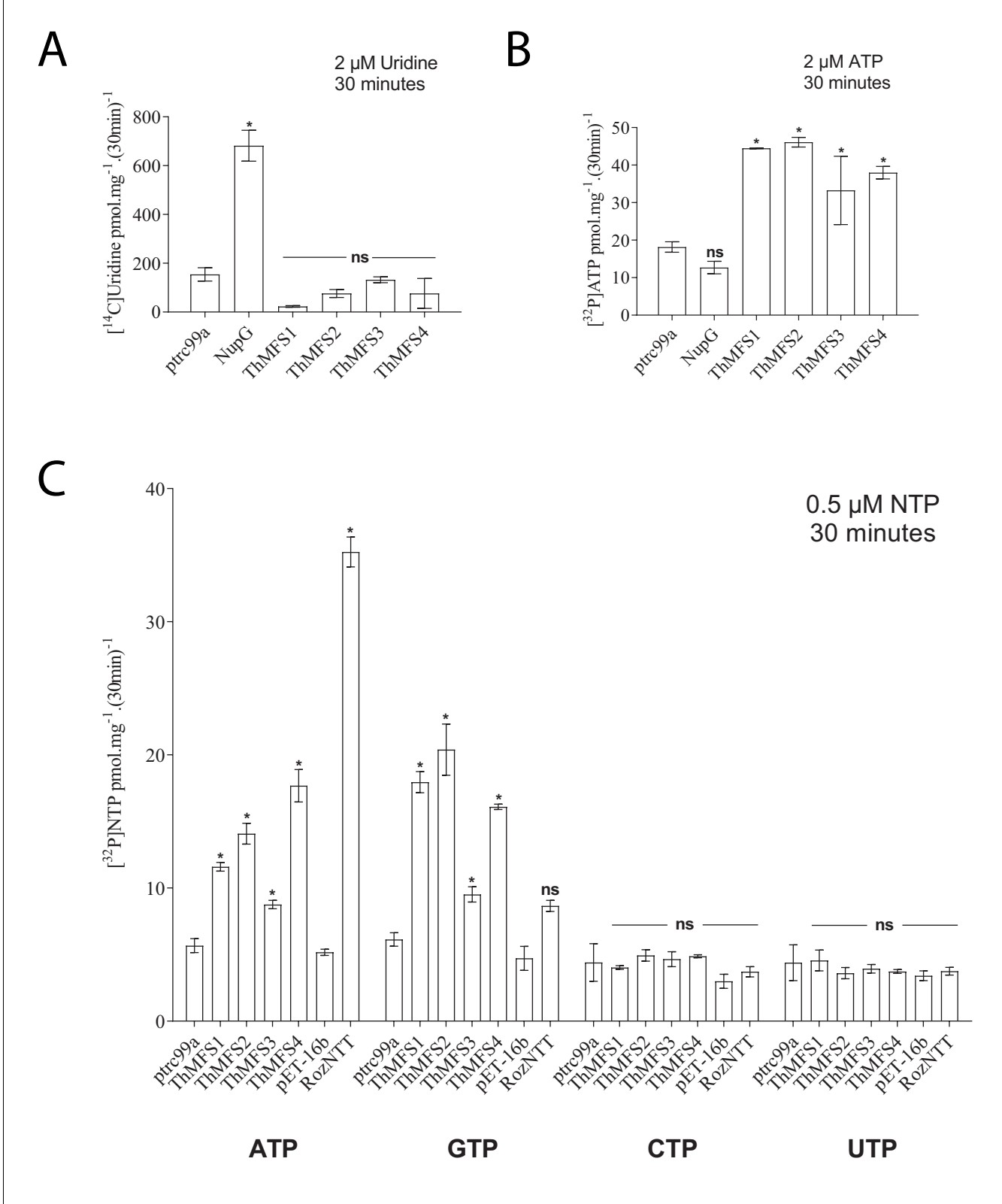

**Figure 3.** Transport assay for the nucleoside uridine and selected nucleotides in *E. coli* expressing recombinant *E. coli* NupG, ThMFS1-4 proteins or *Rozella allomycis* NTT. (**A**) Radiolabelled uridine uptake assay with *E. coli* cells expressing the native *E. coli* NupG transporter or one of the four ThMFS proteins (ThMFS1-4) cloned into the expression vector ptrc99a. The empty ptrc99a plasmid was used as a control for background transport of the radiolabelled substrate. (**B**) Radiolabelled ATP import assay for the same five genes as in (**A**) and with the same control of the empty ptrc99a plasmid.
*Figure 3 continued on next page*

Figure 3 continued

(C) Uptake assays for the four radiolabelled nucleotides ($\alpha^{32}$P-) ATP, GTP, CTP and UTP using the same expression system as in A and B. The same control (empty ptrc99a plasmid) was used for each tested substrate. For the *Rozella* NTT (RozNTT) cloned in pET16b, the control was the empty plasmid pET16b (**Dean et al., 2018**). N = 3 for each condition and the error bars represent standard deviations. Significant differences at $p < 0.05$ (one-way ANOVA) between controls (empty plasmids, ptrc99a or pET-16b) and individual transporters are shown with * (ns: non-significant).
DOI: https://doi.org/10.7554/eLife.47037.023

The following source data is available for figure 3:

**Source data 1.** Raw data and their processing to calculate the transport of the tested nucleoside or nucleotides by ThMFS1-4 and control transporters expressed in *E. coli* (worksheet 1: panels A and B; worksheet 2: panel C).
DOI: https://doi.org/10.7554/eLife.47037.024

(**Figure 2C**). The functional relevance of the transporters ThMFS2 and ThMFS4 is unclear as we were unable to detect any evidence of their protein expression in our model system.

It was originally suggested that the *Nematocida* homologues of the *T. hominis* ThMFS transporters might have similar transport properties to the *E. coli* MFS NupG (**Cuomo et al., 2012**). NupG is an MFS H$^+$ symporter that uses the proton gradient to drive transport of purine and pyrimidine nucleosides (**Xie et al., 2004**). However, as we show in the present study, the levels of shared sequence similarity between NupG and Microsporidia sequences are generally low. Moreover, characterised members of the MFS superfamily that includes the Microsporidia sequences and NupG, transport a broad spectrum of ions and solutes using a variety of different mechanisms (**Yan, 2015**). This diversity makes it difficult to make reliable functional inferences based upon sequence similarity (**Finn et al., 2016**; **Saier et al., 2016**; **Yan, 2015**). This is exemplified by recently published data (**Dean et al., 2018**) for Microsporidia NTT transporters where substrate and mechanism vary among closely related paralogues (**Dean et al., 2018**). In this study, we show that ThMFS1-4 do not transport uridine, a known pyrimidine nucleoside substrate of NupG (**Xie et al., 2004**), and that ThMFS1 and ThMFS3, are not inhibited by CCCP, a classic inhibitor of H$^+$ symporters like NupG (**Xie et al., 2004**). Moreover, although *T. hominis* has a putative uridine kinase (**Heinz et al., 2012**), most Microsporidia genomes appear to lack the kinases needed to utilise uridine and other nucleosides in their metabolism (**Dean et al., 2016**), even if they could import them.

In recent years, a number of endoparasitic microbial eukaryotes related to Microsporidia have been identified and their genomes sequenced (**Galindo et al., 2018**; **Haag et al., 2014**; **James et al., 2013**; **Mikhailov et al., 2017**; **Quandt et al., 2017**). Phylogenomic analyses suggest that these taxa form a monophyletic sister group to the true fungi, for which the name Rozellomycota has been suggested (**Corsaro et al., 2016**), and that core Microsporidia form a distinct clade within Rozellomycota (**Galindo et al., 2018**; **Mikhailov et al., 2017**; **Quandt et al., 2017**). Like the Microsporidia, many of the Rozellomycota lack genes for making nucleotides de novo and some have a limited capacity for making their own ATP (**Dean et al., 2016**; **Dean et al., 2018**; **Galindo et al., 2018**; **James et al., 2013**; **Mikhailov et al., 2017**; **Quandt et al., 2017**). This suggests that they all depend to some degree upon host energy and nucleotides for their own intracellular growth and replication. Recent work has shown that *Rozella* and Microsporidia possess horizontally acquired nucleotide (NTT) transport proteins that they can use to import host ATP, and in the case of Microsporidia other purine nucleotides and NAD$^+$ (**Dean et al., 2018**). The topology of the NTT tree (**Dean et al., 2018**) suggests that NTT transporters were acquired by lateral gene transfer into the common ancestor of Rozella and Microsporidia and subsequently vertically inherited (**Dean et al., 2018**; **Major et al., 2017**). Analysis of the draft genomes of *Amphiamblys* sp. (**Mikhailov et al., 2017**), *Metchnikovella incurvata* (**Galindo et al., 2018**), *Mitosporidium daphnia* (**Haag et al., 2014**) and *Paramicrosporidium saccamoebae* (**Quandt et al., 2017**), suggest that they lack NTT genes. Based upon the topology of published trees (**Dean et al., 2018**; **Galindo et al., 2018**; **James et al., 2013**; **Mikhailov et al., 2017**; **Quandt et al., 2017**) which place these species as internal branches of the Rozellomycota, it appears that they have lost NTT genes during their independent evolution. By contrast, all these species contain multiple homologues of the ThMFS transporters (**Figure 1**, **Figure 1—source data 2**) suggesting that these transporters provide an alternative and hitherto unrecognised way for endoparasites to exploit the host nucleotide pool.

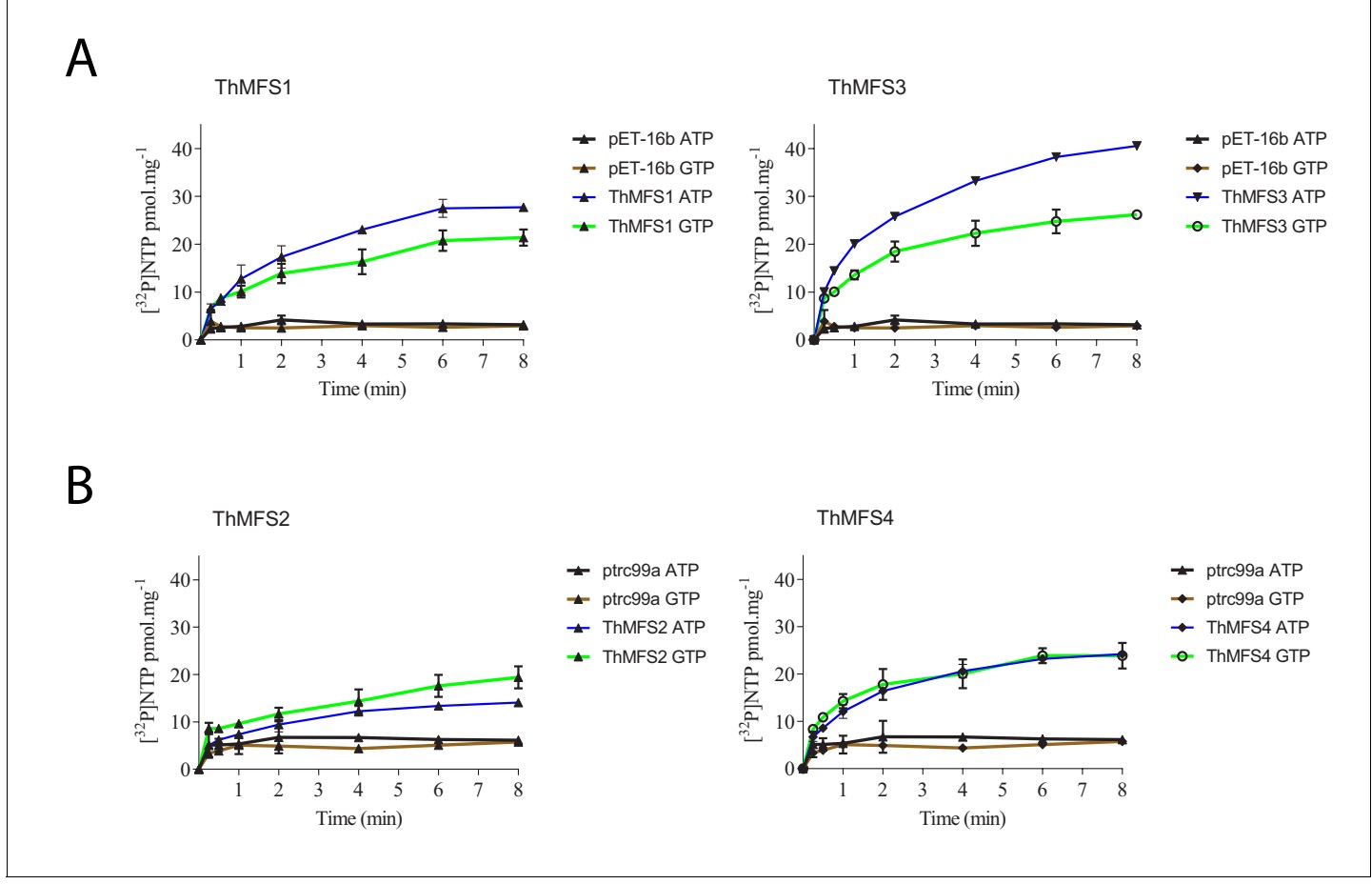

**Figure 4.** Time course of ATP and GTP uptake by *E. coli* cells expressing recombinant ThMFS1-4 proteins. Each ThMFS transporter was assayed using both *E. coli*-expression vector systems (pET16b or ptrc99a) with the results shown being taken from the experiment with the highest transport activity. In each experiment, the corresponding empty plasmid was used as control for background transport. The indicated substrates were all used at 0.5 µM. (A) Uptake assay for the ThMFS1 (THOM_0963) protein expressed using pET16b in *E. coli* Rosetta2(DE3)pLysS and the native (not codon optimised) ORF. (B) Uptake assay for the ThMFS2 (THOM_1192) protein expressed with the ptrc99a plasmid system in *E. coli* GD1333 and the *E. coli* codon optimised synthetic ORF. (C) Uptake assay for the ThMFS3 (THOM_1681) protein expressed with pET16b system in *E. coli* Rosetta2(DE3)pLysS and the native ORF. (D) Uptake assay for the ThMFS4 (THOM_3170) protein expressed with the ptrc99a plasmid system in *E. coli* GD1333 and the *E. coli* codon optimised synthetic ORF. N = 3 for each condition with the error bars representing standard deviations. All 8 min time points for specified transporters and nucleotides were significantly different at $p<0.05$ (one-way ANOVA) from controls (empty plasmids, ptrc99a or pET-16b).

DOI: https://doi.org/10.7554/eLife.47037.025

The following source data is available for figure 4:

**Source data 1.** Raw data and their processing to calculate the transport of the tested nucleotides by ThMFS1-4 expressed in *E. coli* (worksheet 1: panels A and C; worksheet 2: panels B and D).

DOI: https://doi.org/10.7554/eLife.47037.026

# Materials and methods

## Key resources table

| Reagent type (species) or resource | Designation | Source or reference | Identifiers | Additional information |
|---|---|---|---|---|
| Gene (*T. hominis*) | ThMFS1, native ORF, MFS family c_456 in *Nakjang et al. (2013)* | PCR cloned *T. hominis* gDNA *Heinz et al. (2012)* | JH993885.1 (GenBank accessions) | THOM_0963 (locus tags) |

*Continued on next page*

*Continued*

| Reagent type (species) or resource | Designation | Source or reference | Identifiers | Additional information |
|---|---|---|---|---|
| Gene (*T. hominis*) | ThMFS2, native ORF MFS family c_456 in *Nakjang et al. (2013)* | PCR cloned *T. hominis* gDNA *Heinz et al. (2012)* | MH824667 (GenBank accessions) | THOM_1192 (locus tags) |
| Gene (*T. hominis*) | ThMFS3, native ORF, MFS family c_456 in *Nakjang et al. (2013)* | PCR cloned *T. hominis* gDNA *Heinz et al. (2012)* | MH824668 (GenBank accessions) | THOM_1681 (locus tags) |
| Gene (*T. hominis*) | ThMFS4, native ORF, MFS family c_456 in *Nakjang et al. (2013)* | PCR cloned *T. hominis* gDNA *Heinz et al. (2012)* | JH994098.1 (GenBank accessions) | THOM_3170 (locus tags) |
| Gene (*T. hominis*) | ThMFS1 Synthetic ORF | GeneArt, Thermo Fisher Scientific | MH824663 (GenBank accessions) | Codon optimised for *E. coli* with an HA tag at the C-termini |
| Gene (*T. hominis*) | ThMFS2 Synthetic ORF | GeneArt, Thermo Fisher Scientific | MH824664 (GenBank accessions) | Codon optimised for *E. coli* with an HA tag at the C-termini |
| Gene (*T. hominis*) | ThMFS3 Synthetic ORF | GeneArt, Thermo Fisher Scientific | MH824665 (GenBank accessions) | Codon optimised for *E. coli* with an HA tag at the C-termini |
| Gene (*T. hominis*) | ThMFS4 Synthetic ORF | GeneArt, Thermo Fisher Scientific | MH824666 (GenBank accessions) | Codon optimised for *E. coli* with an HA tag at the C-termini |
| Sequenced-based reagent | THOM_0963 VspI F | This study | PCR primer ThMFS1-F | TGCACCATTAATGAACCGTT TTTGAACATG |
| Sequenced-based reagent | THOM_0963 BglII R | This study | PCR primer ThMFS1-R | CACTTGAGAT CTTTACATCG TAGACTTAGG |
| Sequenced-based reagent | THOM_1192 NdeI F | This study | PCR primer ThMFS2-F | TGCACCCATA TGCCATCAAT GAATAGGTCC |
| Sequenced-based reagent | THOM_1192 BamHI R | This study | PCR primer ThMFS2-R | CACTTGGGAT CCTTATTTGT TCCTCTTTTT |
| Sequenced-based reagent | THOM_1681 NdeI F | This study | PCR primer ThMFS3-F | TGCACCCATA TGGATTGCCG GCTTTTGAGT |
| Sequenced-based reagent | THOM_1681 BamHI R | This study | PCR primer ThMFS3-R | CACTTGGGAT CCTCACTCAA TTTCCGCAGG |
| Sequenced-based reagent | THOM_3170 NdeI F | This study | PCR primer ThMFS4-F | TGCACCCATA TGCACAGAAA TTTTATACTC |
| Sequenced-based reagent | THOM_3170 BamHI R | This study | PCR primer ThMFS4-R | CACTTGGGAT CCTTATTTGT GTGCGGTCCA |
| Sequenced-based reagent | NupG XbaI F | This study | PCR primer NupG-F | TGCACCTCTA GAATGAATCTT AAGCTGCAG |
| Sequenced-based reagent | NupG HindIII R | This study | PCR primer NupG-R | CACTTGAAGC TTTTAGTGGCT AACCGTCTG |
| Cell line (*Oryctolagus cuniculus*) | Rabbit kidney cell line RK13 | LGC-Standards – ATCC | ATCC CCL-37 | Infected with the Bovine Viral Diarrhea Virus |

*Continued on next page*

*Continued*

| Reagent type (species) or resource | Designation | Source or reference | Identifiers | Additional information |
|---|---|---|---|---|
| Strain, strain background (*E. coli*) | *E. coli* GD1333 | **Nørholm and Dandanell (2001)** | Kindly provided by Prof. Gert Dandanell, University of Copenhagen | This strain does not express the two native nucleoside transporters NupG and NupC |
| Strain, strain background (*E. coli*) | *E. coli* Rosetta2 (DE3) pLysS | Novagen | | BL21 derivative designed to enhance the expression of eukaryotic proteins that contain codons rarely used in *E. coli* |
| Recombinant DNA reagent | pET16b-ThMFS1 | This study | | Native ORF, PCR cloned (JH993885.1) |
| Recombinant DNA reagent | pET16b-ThMFS2 | This study | | Native ORF, PCR cloned (MH824667) |
| Recombinant DNA reagent | pET16b-ThMFS3 | This study | | Native ORF, PCR cloned (MH824668) |
| Recombinant DNA reagent | pET16b-ThMFS4 | This study | | Native ORF, PCR cloned (JH994098.1) |
| Recombinant DNA reagent | ptrc99a-ThMFS1 | This study | | Synthetic ORF with codon optimised for *E. coli* (MH824663) XbaI – HindIII |
| Sequenced-based reagent | ptrc99a-ThMFS2 | This study | | Synthetic ORF codon optimised for *E. coli* (MH824664) XbaI – HindIII |
| Sequenced-based reagent | ptrc99a-ThMFS3 | This study | | Synthetic ORF with codon optimised for *E. coli* (MH824665) XbaI – HindIII |
| Sequenced-based reagent | ptrc99a-ThMFS4 | This study | | Synthetic ORF with codon optimised for *E. coli* (MH824666) XbaI – HindIII |
| Sequenced-based reagent | ptrc99a-NupG | This study | Rosetta2(DE3) pLysS gDNA for PCR cloning | Native *E. coli* NupG ORF, PCR cloned |
| Sequenced-based reagent | pET16b-RozNTT | **Dean et al. (2018)** | | Native *Rozella allomycis* NTT ORF, PCR cloned |
| Sequenced-based reagent | pET16b-PamNTT5 | **Haferkamp et al. (2006)** | Kindly provided by Dr Ilka Haferkamp, University of Kaiserslautern | Native *Protochlamydia amoebophila* NTT5 ORF, PCR cloned |
| Antibody | anti-ThMFS1 peptides (affinity purified, rabbit polyclonal) | BioGenes GmbH (Germany). This study | Peptides: CIKSYDRAER SNADIES and CEDEGDNKPS NPKST | IFA: 1:50; WB: 1:1000 |

*Continued on next page*

*Continued*

| Reagent type (species) or resource | Designation | Source or reference | Identifiers | Additional information |
|---|---|---|---|---|
| Antibody | anti-ThMFS2 peptides (affinity purified, rabbit polyclonal) | BioGenes GmbH (Germany). This study | Peptides: CKTPKFKKDV KENLTREGR and CIDRDLKDPR TVNEDES | IFA: 1:2, 1:10, or 1:50; WB: 1:1000 |
| Antibody | anti-ThMFS3 peptides (affinity purified, rabbit polyclonal) | BioGenes GmbH (Germany). This study | Peptides: CNYLEHEGLD VRQSGR and CFSRRLRGEG TKNREN | IFA: 1:50; WB: 1:1000 |
| Antibody | anti-ThMFS4 peptides (affinity purified, rabbit polyclonal) | BioGenes GmbH (Germany). This study | Peptides: CVKRTNSSNR NVGTAK and CKPEAVLFKR KISLKD | IFA: 1:2, 1:10, or 1:50; WB: 1:1000 |
| Antibody | anti-ThmitHsp70 (rat, polyclonal) | *Heinz et al. (2014)*; *Dean et al. (2018)* | | IFA: 1:200 |
| Antibody | Alexa Fluor 488 Goat Anti-Rat IgG (H+L) (polyclonal) | ThermoFisher Scientific | | IFA: 1:500 |
| Antibody | Alexa Fluor 594 Goat Anti-Rabbit IgG (H+L) (polyclonal) | ThermoFisher Scientific | | IFA: 1:500 |
| Antibody | Peroxidase (HRP) conjugated Goat Anti-Rabbit IgG Antibody (polyclonal) | Sigma | | WB: 1:10000 |

## Bioinformatic and phylogenetic analyses

Homologues of the Microsporidia proteins from the MFS family c_456 in *Nakjang et al. (2013)* were collected by performing BlastP searches against the NCBI non-redundant protein (nr) database. *Escherichia coli* NupG and annotated (PF03825) sequences at the Pfam database (*Finn et al., 2016*), as well as selected sequences from previously published phylogenies (*James et al., 2013*; *Nakjang et al., 2013*; *Xie et al., 2004*) served as initial queries. A HMMer search based upon the sampled MFS proteins was performed against the Pfam profiles using the Pfam server (*Finn et al., 2016*). For profile-profile searches we took advantage of the HHPred server (*Söding et al., 2005*).

Alignments for sequence comparisons and phylogenetic analyses were generated within SEA-VIEW (*Gouy et al., 2010*) using MUSCLE (*Edgar, 2004*) and trimmed with TrimAl using the setting 'gappyout' (*Capella-Gutiérrez et al., 2009*) leading to a total of 477 aligned residues for the 63 species Rozellomycota dataset and 464 residues for the broader taxonomic dataset. Phylogenies were inferred with Maximum likelihood using the LG+C60 model in IQ-TREE (*Nguyen et al., 2015*) with 1000 ultrafast bootstrap replicates (*Hoang et al., 2018*).

## Chemicals

[U-$^{14}$C] uridine (539 mCi/mmol) was purchased from Perkin Elmer. $\alpha^{32}$P-labelled ATP GTP, CTP, and UTP (3000 Ci/mmol or 800 Ci/mmol) were obtained from Hartmann Analytic or Perkin Elmer. Cold nucleotides and nucleosides were obtained from Sigma Aldrich and prepared according to the manufacturer's instructions. The protonophore CCCP was from Sigma Aldrich.

## Cultivation of *Trachipleistophora hominis*

Microsporidia were grown in rabbit kidney (RK13) host cells at 33°C in complete DMEM as described previously (*Dean et al., 2018*; *Goldberg et al., 2008*; *Heinz et al., 2014*; *Heinz et al., 2012*). The RK13 rabbit kidney cell line is positive for the bovine viral diarrhea virus (BVDV), obtained from the ATCC: ATCC CCL-37 and Mycoplasma free. The parasite *T. hominis* was originally obtained from Prof. Liz Canning (*Williams et al., 2002*).

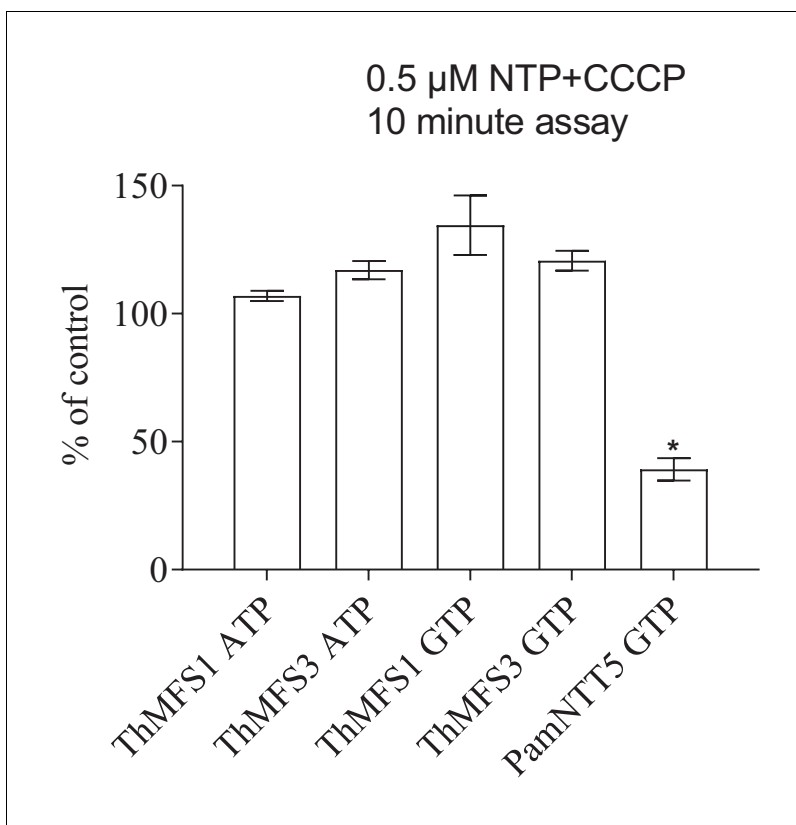

**Figure 5.** Lack of impact of the protonophore CCCP on nucleotide import in *E. coli* expressing the two parasite cell-surface located ThMFS proteins. [$\alpha^{32}$P]-nucleotide import by ThMFS1, ThMFS3, and by the *Protochlamydia amoebophila* symporter NTT5 (*Pam*NTT5) (*Haferkamp et al., 2006*) was compared in the absence (control, set to 100%) and presence of the protonophore CCCP. The GTP and ATP H$^+$-symporter PamNTT5 was used as a positive control for CCCP inhibition (*Haferkamp et al., 2006*). N = 3 for each condition with the error bars representing standard deviations. The significant reduction of transport between control (no CCCP) and CCCP treatments at $p < 0.05$ (one-way ANOVA) is indicated with *.

DOI: https://doi.org/10.7554/eLife.47037.027

The following source data and figure supplement are available for figure 5:

**Source data 1.** Raw transport data and their processing to investigate the impact of the protonophore CCCP on nucleotide import in *E. coli* expressing ThMFS1, ThMFS3 or the control PamNTT5 (*Figure 5*, *Figure 5—figure supplement 1*).

DOI: https://doi.org/10.7554/eLife.47037.029

**Figure supplement 1.** Impact of the protonophore CCCP on nucleotide transport by *E. coli* expressing ThMFS1, ThMFS3 or the control PamNTT5.

DOI: https://doi.org/10.7554/eLife.47037.028

## RNA-Seq data generation and analyses

To initiate the synchronised infection by *T. hominis*, non-infected RK13 cells grown on 150 cm$^2$ round tissue culture dishes were incubated with freshly purified spores as previously described (*Watson et al., 2015*) before processing for RNA-Seq analysis as described (*Dean et al., 2018*; *Watson et al., 2015*). Briefly, 2 hr after the addition of *T. hominis* spores to the RK13 cells monolayer, the host cell monolayer was extensively washed with PBS in order to remove spores, followed by the addition of fresh culture medium. Two tissue culture dishes were processed independently for RNA purification, sequencing and mRNA quantification (*Watson et al., 2015*) at six different time points post-infection: 3 hr, 14 hr, 22 hr, 40 hr, 70 hr and 96 hr (*Dean et al., 2018*). Previous studies (in triplicate) indicated that the variation between independent replicates for this experimental design was low (*Watson et al., 2015*). Library preparation and sequencing were carried out using the Illumina stranded preparation kit and run on two lanes of an Illumina HiSeq 2500, which

produced 27.5 M paired-end reads from both the RK13 host cells and *T. hominis* transcripts. CutAdapt (*Martin, 2011*) was used to remove adapter sequences and low quality 3′ sequence regions (set at the q −20 threshold). Two approaches were used to quantify transcripts in the resulting dataset. Trimmed reads were mapped to the *T. hominis* genome using TopHat2 (*Kim et al., 2013*) and transcripts assembled and quantified using Cufflinks (*Trapnell et al., 2012*) as previously described (*Dean et al., 2018*; *Watson et al., 2015*). Transcript abundances were presented in FPKM (Fragments Per Kilobase per Million mapped reads) (*Figure 2—source data 2*) (*Dean et al., 2018*; *Watson et al., 2015*). In a second approach (presented in *Figure 2A*, *Figure 2—source data 2*), predicted *T. hominis* transcripts were quantified by pseudoalignment of reads using kallisto (*Bray et al., 2016*). The results were then analysed using sleuth (*Pimentel et al., 2017*) with transcript abundances presented as Transcripts Per Kilobase per million mapped reads (TPM) (*Wagner et al., 2012*). The RNA-Seq data were submitted to GenBank and listed at the NCBI under the existing BioProject PRJNA278775 with the BioSample accession numbers SAMN11265032-SAMN11265043 (one accession number for each of the two samples per time point post infection).

## Antisera generation

Pairs of peptides were selected from hydrophilic segments of the ThMFS1-4 amino acid sequences and used to generate custom-made antisera targeting each of the four ThMFS1-4 candidate transporters (*Figure 2—figure supplement 6*). Anti-peptides antisera were produced in rabbits by BioGenes GmbH (Germany) and commercially affinity purified against the peptide antigens. Two rabbits for each pair of peptides were processed in parallel. To test the antisera specificity for their respective peptides, we used western blots with 300 ng of peptide directly spotted onto nitrocellulose membranes, affinity purified antibodies were diluted 1:1000 and secondary goat anti-rabbit antisera conjugated to HRP were diluted 1:10000 (*Figure 2—figure supplement 6*). Western blot detection was processed with a chemiluminescent substrate (Thermo Scientific - Pierce ECL) following the manufacturer's instructions. Image development was processed with a Biorad gel imager ChemiDoc XRS+ with images taken every 10 s.

## Immunofluorescent microscopy

The same *T. hominis* infected RK13 cells used for the synchronised infection experiments (see RNA-Seq section) were also grown on 13 mm glass coverslips and then fixed in methanol/acetone (50:50 at −20°C for at least 10 min) (*Dean et al., 2018*; *Goldberg et al., 2008*; *Heinz et al., 2014*; *Tsaousis et al., 2008*) at the different time points post-infection. Rat antisera against ThmtHSP70 (locus tag THOM_3057) were used to label *T. hominis* mitosomes and both host and parasite nuclei were stained with DAPI, as previously described (*Dean et al., 2018*; *Freibert et al., 2017*; *Goldberg et al., 2008*; *Heinz et al., 2014*; *Tsaousis et al., 2008*). Dilutions of antisera were as follows: anti-ThmtHSP70 1:200; anti-ThMFS1-4 1:50. For anti-ThMFS2 and anti-ThMFS4 1:10 and 1:2 dilutions were also tested but no IFA signal was observed in any of the tested conditions. For ThMFS1 and ThMFS3 both rabbit antisera gave parasite-specific IFA signals. Blocking and antibody incubations were performed in 5% milk prepared from skimmed milk powder in PBS, with PBS used for washing steps. Cells were stained with DAPI (Molecular Probes) and mounted in Vectashield Hard set (VectorLabs). Peptide competition assays were performed with the addition of solubilized pairs of peptides in a 200-fold molar excess compared to the affinity-purified antibody. The IgG molar mass was considered as 150,000 Daltons, and the molar masses of the blocking peptides were taken into consideration for the calculations. For ThMFS1, the final antibody concentration was 5.2 µg/ml and the concentrations of peptide 1 and peptide 2 were 13.6 µg/ml and 11.2 µg/ml, respectively. For ThMFS3, the final antibody concentration was 3.3 µg/ml and the concentrations of peptide 1 and peptide 2 were 8.3 µg/ml and 8.5 µg/ml, respectively. As expected, the control mitosomal ThmitHsp70 signal was not affected by any of these conditions (*Figure 2—figure supplement 9*). Images were taken with a Zeiss Axioimager II epifluorescence microscope using a 63x objective lens and images processed with ImageJ (*Schneider et al., 2012*). Quantification of the ThMFS antibody fluorescence signal was determined for fields including intracellular stages of the parasite. Axiovision software and Image J[Fiji] with the Bioimport plugin was used to determine relative

fluorescence for each specific antibody including the mitosome-specific antibody to mitHsp70. Background signal was determined in a similar manner and subtracted from the specific signal.

## Western blot analyses on total protein extracts from cell cultures and purified spores

For total protein extracts, confluent monolayers of either *T. hominis*-infected or non-infected RK13 cells were washed three times with PBS, and lysed with ice cold 2% SDS-PBS lysis buffer containing protease inhibitor cocktail (Sigma P8340), PMSF (Sigma), MgCl2 (0.6 mM final concentration), and Benzonase (25U, Novagen). *T. hominis* spores were purified from the culture medium from *T. hominis* infected RK13 cultures using Percoll (Sigma) density gradient centrifugation. Purified spores were lysed by boiling for 10 min in 2% SDS-PBS lysis buffer with protease inhibitors. The protein concentration of the samples for Western blotting was determined with a BCA assay (Pierce BCA Protein Assay Kit). The samples were boiled for 5 min in Laemmli loading buffer and the indicated amount of total protein extracts were loaded per lane for SDS–PAGE. To estimate the size of the detected proteins and to monitor protein migration and transfer we used the PageRuler prestained protein ladder, 10 to 180 kDa (ThermoFisher Scientific). Western blot analysis used the indicated affinity purified rabbit antibodies (diluted 1:1000) in combination with HRP-conjugated secondary goat anti-rabbit antisera (diluted 1:10000) (Sigma). Image development was processed using a Biorad gel imager ChemiDoc XRS+.

## Heterologous expression and transport assays in *E. coli* cells

The four native ThMFS genes were PCR amplified from purified genomic DNA and cloned into the pET-16b expression vector. Sequences of cloned genes were verified and submitted to GenBank when distinct from the previously deposited genome sequence-derived ORF (*Heinz et al., 2012*). These have the following accession numbers: ThMFS2_native: MH824667, ThMFS3_native: MH824668. Codon-optimized genes for expression in *E. coli* that contained a C-terminal 2x HA-tag were produced synthetically (GeneArt, Thermo Fisher Scientific) and then cloned into the ptrc99a plasmid. These sequences have the following GenBank accession numbers: ThMFS1_synthetic: MH824663; ThMFS2_synthetic: MH824664; ThMFS3_synthetic: MH824665; ThMFS4_synthetic: MH824666. Proteins were expressed as described previously for Microsporidian NTTs (*Dean et al., 2018*; *Heinz et al., 2014*; *Tsaousis et al., 2008*). T7-based constructs were used for expression in *E. coli* Rosetta2(DE3)pLysS and trc-based constructs were expressed in *E. coli* GD1333 (kindly provided by Prof. Gert Dandanell) (*Nørholm and Dandanell, 2001*). Briefly, a starting culture from a single colony was grown in LB at 37°C overnight and used to inoculate a 50 ml culture grown at 37°C until an $OD_{600}$ of 0.4–0.6. Protein expression was induced by addition of 1 mM IPTG and the culture was incubated for and additional 16 hr at 18°C. Cells were harvested at 5000 x *g* for 10 min and were washed twice with PBS to remove residual medium. The cells were re-suspended in PBS with the $OD_{600}$ adjusted to 5/ml. Cultures were kept in PBS at room temperature until used for uptake assays. Transport of nucleosides and nucleotides was assayed using 0.5 µM or 2 µM of radiolabelled substrate, as described in the main text/Figures. $\alpha^{32}P$-radioisotopes were present at 1–2 µCi/ml and used as described (*Audia and Winkler, 2006*), and [U-$^{14}C$] isotopes were used at 2.5–5 mCi/mmol as described (*Nørholm and Dandanell, 2001*). To test if a proton gradient affected transport by *E. coli* cell expressing the ThMFS1 and ThMFS3 transporters, we added 250 µM of CCCP to dissipate the $H^+$ gradient across the bacterial membranes prior to transport assays (*Dean et al., 2018*; *Haferkamp et al., 2006*).

## Acknowledgements

We thank Ekaterina Kozhevnikova and Maxine Geggie for assistance with tissue culture and general technical support. The *E. coli* strain GD1333 was kindly provided by Prof. Gert Dandanell. We thank the referees for providing constructive feedback which allowed us to improve the manuscript.

## Additional information

### Funding

| Funder | Grant reference number | Author |
|---|---|---|
| Wellcome | 089803/Z/09/Z | T Martin Embley |
| European Research Council | Advanced Investigator Program (ERC 2010-268701) | T Martin Embley |
| Biotechnology and Biological Sciences Research Council | PhD studentship | Andrew K Watson |

The funders had no role in study design, data collection and interpretation, or the decision to submit the work for publication.

### Author contributions

Peter Major, Investigation, Writing—original draft; Kacper M Sendra, Paul Dean, Investigation; Tom A Williams, Formal analysis; Andrew K Watson, Formal analysis, Investigation; David T Thwaites, Conceptualization, Supervision, Writing—review and editing; T Martin Embley, Robert P Hirt, Conceptualization, Supervision, Funding acquisition, Writing—original draft, Project administration, Writing—review and editing

### Author ORCIDs

Peter Major (iD) https://orcid.org/0000-0002-6443-5631
T Martin Embley (iD) https://orcid.org/0000-0002-1484-340X
Robert P Hirt (iD) https://orcid.org/0000-0002-3760-9958

### Decision letter and Author response

Decision letter https://doi.org/10.7554/eLife.47037.050
Author response https://doi.org/10.7554/eLife.47037.051

## Additional files

### Supplementary files

• Transparent reporting form
DOI: https://doi.org/10.7554/eLife.47037.030

### Data availability

New sequences data were submitted to GenBank: 1) RNA-Seq data: BioProject PRJNA278775 with the BioSample accession numbers SAMN11265032-SAMN11265043 (one accession for each of the two samples per time point post infection); 2) The new native PCR cloned gene sequences have the following GenBank accession numbers: ThMFS2_native: MH824667, ThMFS3_native: MH824668; 3) Codon-optimized genes for expression in *E. coli* have the following GenBank accession numbers: ThMFS1_synthetic: MH824663, ThMFS2_synthetic: MH824664, ThMFS3_synthetic: MH824665, ThMFS4_synthetic: MH824666. These are all listed in the Materials and Methods section, see Key Resources Table.

The following datasets were generated:

| Author(s) | Year | Dataset title | Dataset URL | Database and Identifier |
|---|---|---|---|---|
| Watson AK, Sendra KM, Dean P, Williams TA, Hirt RP, Embley TM | 2019 | Transcriptomic profiling of host-parasite interactions in the microsporidian Trachipleistophora hominis | https://www.ncbi.nlm.nih.gov/bioproject/?term=PRJNA278775 | NCBI BioProject, PRJNA278775 |
| Major P, Watson AK, Sendra KM, Dean P, Williams TA, Hirt RP, Emb- | 2019 | New native PCR cloned gene sequence (ThMFS2_native) | https://www.ncbi.nlm.nih.gov/nuccore/MH824667 | NCBI GenBank, MH824667 |

ley TM

| Major P, Watson AK, Sendra KM, Dean P, Williams TA, Hirt RP, Embley TM | 2019 | New native PCR cloned gene sequence (ThMFS3_native) | https://www.ncbi.nlm. nih.gov/nuccore/ MH824668 | NCBI GenBank, MH824668 |
|---|---|---|---|---|
| Major P, Watson AK, Sendra KM, Dean P, Williams TA, Hirt RP, Embley TM | 2019 | Codon-optimized genes for expression in *E. coli* (ThMFS1_synthetic) | https://www.ncbi.nlm. nih.gov/nuccore/ MH824663 | NCBI GenBank, MH824663 |
| Major P, Watson AK, Sendra KM, Dean P, Williams TA, Hirt RP, Embley TM | 2019 | Codon-optimized genes for expression in *E. coli* (ThMFS2_synthetic) | https://www.ncbi.nlm. nih.gov/nuccore/ MH824664 | NCBI GenBank, MH824664 |
| Major P, Watson AK, Sendra KM, Dean P, Williams TA, Hirt RP, Embley TM | 2019 | Codon-optimized genes for expression in *E. coli* (ThMFS3_synthetic) | https://www.ncbi.nlm. nih.gov/nuccore/ MH824665 | NCBI GenBank, MH824665 |
| Major P, Watson AK, Sendra KM, Dean P, Williams TA, Hirt RP, Embley TM | 2019 | Codon-optimized genes for expression in *E. coli* (ThMFS4_synthetic) | https://www.ncbi.nlm. nih.gov/nuccore/ MH824666 | NCBI GenBank, MH824666 |

The following previously published datasets were used:

| Author(s) | Year | Dataset title | Dataset URL | Database and Identifier |
|---|---|---|---|---|
| Heinz E, Williams TA, Nakjang S, Noël CJ, Swan DC, Goldberg AV, Harris SR, Weinmaier T, Markert S, Becher D, Bernhardt J, Dagan T, Hacker C, Lucocq JM, Schweder T, Rattei T, Hall N, Hirt RP, Embley TM | 2012 | Annotation of genome data | https://www.ncbi.nlm. nih.gov/bioproject/84343 | NCBI Bioproject, PRJNA84343 |
| Watson AK, Williams TA, Williams BA, Moore KA, Hirt RP, Embley T | 2015 | RNA-Seq | https://www.ncbi.nlm. nih.gov/bioproject/ 278775 | NCBI Bioproject, PRJNA278775 |

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
