## [Decision Letter]

Thank you for submitting your article "A new family of cell surface located purine transporters in Microsporidia and related fungal endoparasites" for consideration by *eLife*. Your article has been reviewed by three peer reviewers, including Emily Troemel as the Reviewing Editor and Reviewer #1, and the evaluation has been overseen by Anna Akhmanova as the Senior Editor. The following individual involved in review of your submission has agreed to reveal their identity: Gira Bhabha (Reviewer #3).

The reviewers have discussed the reviews with one another and the Reviewing Editor has drafted this decision to help you prepare a revised submission.

Summary:

Microsporidia are obligate intracellular parasites that can infect a broad range of animal hosts. They have lost several biosynthetic pathways such as those for de novo synthesis of nucleotides and thus appear to rely exclusively on their hosts for nucleotides. Previous work had characterized a family of purine nucleotide transporters in microsporidia called NTTs, which appear to steal purine nucleotides from host cells. In this paper, the authors characterize a gene family present in all microsporidia species that share a Pfam domain with an *E. coli* purine and pyrimidine nucleoside transporter called NupG. This family belongs to the Major Facilitator Superfamily (MFS) and had been identified in previous genome analyses. Here the authors perform more extensive phylogenetic analysis, which indicates that these genes encode a class of transporters that may be ancestral among the eukaryotes, and not horizontally acquired as previously suggested. The authors describe expression of these genes based on previous RNAseq studies and focus in particular on four MFS genes called ThMFS1, 2, 3, 4 from the microsporidia species *T. hominis*. RNAseq analysis and antibody staining of *T. hominis* growing in rabbit kidney cells indicates that ThMFS1 and ThMFS3 are expressed while the parasite is growing in host cells, while ThMFS2 and ThMFS4 have much lower or no expression.

Immunohistochemistry indicates that ThMFS1 and ThMFS3 are localized to the cytoplasmic membrane of the parasite, consistent with a transporter function. Interestingly, by expressing these 4 transporter proteins in *E. coli*, the authors show that they are able to transport radiolabelled purine nucleotides, but not pyrimidine nucleotides or the nucleoside uridine. In addition, the authors demonstrate that ThMFS1 and ThMFS3 are likely not H^+^-symporters as their ability to transport GTP and ATP is not inhibited by the protonophore CCCP.

Overall, this paper will make a solid contribution to the field of microsporidia pathogenesis. It combines phylogenetic, immunohistochemical and functional analyses to support the model that microsporidia use ThMFS transporters to steal purine nucleotides from host cells. Because there has been very little functional information for any microsporidia proteins, these studies will provide an important advance, if paired with further controls/analyses as described below.

Essential revisions:

1) Provide more data/controls for antibodies used in immunofluorescence.

A) Quantify results with antibodies against ThMFS1 and ThMFS3 and define the # of replicates (Figure 2C).

B) Show negative results with antibodies against ThMFS2 and ThMFS4.

C) Provide further clarification about whether lack of staining with the antibodies against ThMFS2 and 4 is biologically meaningful, or whether the antibodies are ineffective (subsection “ThMFS transporters are expressed during infection and localize to the cell surface of intracellular parasites”, second paragraph, was confusing on this point).

D) Explain the reason for using antibodies against mitosomes (positive control for antibody staining method?)

E) Desired but not essential: Perform Westerns with antibodies against ThMFS1 and 3 to determine if they recognize proteins of correct sizes

2) Provide more controls, analyses and information for transport assays.

A) Show positive controls for CTP and UTP transport (Figure 3C).

B) Provide statistics to show significance for transport of ATP and GTP by MFS proteins (Figure 3 and 4).

C) Test MFS proteins for transport of nucleosides or explain why this is not feasible. The authors had negative results when they tested ThMFS proteins for transport of the nucleoside uridine (Figure 3A), but they could test other nucleosides as well, especially given that the MFS proteins were originally annotated as nucleoside transporters. Additional motivation for testing other nucleosides is that *T. hominis* appears to have enzymes for converting adenosine to AMP, cytidine to CMP and thymidine to TMP (Heinz et al., 2014). Thus, if the MFS proteins can transport the nucleosides adenosine, cytidine or thymidine, *T. hominis* might be able to convert these nucleosides into nucleotides.

D) Explain how/when ThMFS2 and 4 may be used for purine uptake, given that they appear to have very low expression but are functional in GTP/ATP transport (Figure 3C).

E) For the time course of ATP and GTP uptake (Figure 4), explain the rationale for using two different *E. coli* strains. *S. rosetta*2(DE3)pLysS should contain native NupG – would this be expected to affect the outcome?

3) Provide more information about RNAseq expression data.

A) Explain how expression data correlates between Figure 2A and 2B

- ThMFS4 appears to increase at late timepoints in 2B, but not 2A.

B) Mention in text that NTT4 has much higher RNAseq expression than the MFS proteins.

4) Clarify phylogenetic analysis

A) Figure 1—source data 1 is confusing and doesn't seem to match up with numbers in the text (63 members). The "HMMER_Pfam_HFS" tab and "Sheet 1" tab have the same genes listed multiple times. The "Locus tags profile hits" tab has what looks like a non-redundant list of genes, but has a question highlighted in yellow there – perhaps left by the authors who forgot to proof and remove it. The "Grand Total" says 116. What does this refer to?

B) Consider *Amphiamblys* as a Microsporidian.

---

## [Author Response]

[…] Overall, this paper will make a solid contribution to the field of microsporidia pathogenesis. It combines phylogenetic, immunohistochemical and functional analyses to support the model that microsporidia use ThMFS transporters to steal purine nucleotides from host cells. Because there has been very little functional information for any microsporidia proteins, these studies will provide an important advance, if paired with further controls/analyses as described below.

We would like to thank the referees and editors for the detailed, constructive and positive feedback on our manuscript. We have revised our manuscript in line with the comments and think that this has improved the presentation and the interpretation of our data. However, we have not carried out additional transport assays including testing new substrates. The experiments are technically very challenging and time consuming (explaining the above-mentioned paucity of functional data on Microsporidia proteins) and we respectfully suggest that they are beyond the scope of revision of a strong paper that already contains a lot of new functional data. We have replied to the individual comments as follows.

Essential revisions:1) Provide more data/controls for antibodies used in immunofluorescence.A) Quantify results with antibodies against ThMFS1 and ThMFS3 and define the # of replicates (Figure 2C).

Done, we have now added a new supplementary figure (Figure 2—figure supplement 2) illustrating the quantification of the IFA data for parasites in different fields from two independent experiments, one that corresponds to the same infections used to generate the RNA-Seq data and Figure 2C, plus an additional independent infection. In the new Figure 2—figure supplement 2 we plot the quantification of the ThMFS1 and ThMFS3 (34 cells per time point) signals and the corresponding mitHsp70 signal associated with a number of parasites (N = 5-6 cells per time point) identified in broader fields from the different cover slips (one from which Figure 2C was derived) and one from distinct slides from an independent time course experiment. This complements Figure 2C and quantifies the overall pattern of signal fluctuation for ThMFS1/3 and mitHsp70 during the infection time course, providing a semi-quantitative assessment of the IFA data.

The legend for Figure 2 was modified accordingly.

B) Show negative results with antibodies against ThMFS2 and ThMFS4.

Done, we have generated a new supplementary figure to summarise the IFA data where the mitHsp70 IFA signal is contrasted to the different ThMFS1/3 and ThMFS2/4 IFA signals (new Figure 2—figure supplement 4). This clearly illustrates the lack of any parasite specific IFA signal for the affinity purified antibodies from the ThMFS2/4 antisera.

The main text has been edited accordingly, see the second paragraph of the subsection “ThMFS transporters are expressed during infection and localize to the cell surface of intracellular parasites”, where we refer to the new Figure 2—figure supplement 4 when commenting on the absence of IFA signal for ThMFS2/4. The legend of Figure 2 was also modified accordingly.

C) Provide further clarification about whether lack of staining with the antibodies against ThMFS2 and 4 is biologically meaningful, or whether the antibodies are ineffective (subsection “ThMFS transporters are expressed during infection and localize to the cell surface of intracellular parasites”, second paragraph, was confusing on this point).

Done, we have further clarified this point in the main text. As we are dealing with a negative result it is difficult to identify why this occurred and we wish to avoid overinterpretation, so our manuscript reflects this ambiguity. Both the Western Blot (WB) and IFA data were positive for ThMFS1 and ThMFS3 whereas the corresponding data for ThMFS2 and ThMFS4 were negative for both the WB and IFA assays. The WB and IFA results were consistent with the different level of transcripts observed for the different transporter genes.

We generated new supplementary figures in response to this point and the related comment E) below. The new supplement Figure 2—figure supplement 5 illustrates the WB results (see related point E below) on total protein extracts from control non-infected RK13 cells, *T. hominis* infected cells and *T. hominis* purified spores – see also above response to point B).

The main text has been edited accordingly, see the second paragraph of the subsection “ThMFS transporters are expressed during infection and localize to the cell surface of intracellular parasites”, where we now refer to the new supplement, Figure 2—figure supplement 5. The legend of Figure 2 was also modified accordingly.

D) Explain the reason for using antibodies against mitosomes (positive control for antibody staining method?)

Done, we used the rat anti mitHsp70 antisera for two reasons:

1) It is a well-established antiserum used in previous publications that provides a control to positively identify parasites prior to the spore stage (3-96 hrs). See Dean et al., 2018, and Heinz et al., 2014.

2) This antiserum represents a control for the fixation and permeabilization of the parasites in the infected cells. The mitHsp70 antigen is located in the matrix of mitosomes, which are double membrane-bounded organelles.

The main text has been edited accordingly, see the second paragraph of the subsection “ThMFS transporters are expressed during infection and localize to the cell surface of intracellular parasites” where we specifically state that the mitHsp70 acts as a control for the IFA procedure.

E) Desired but not essential: Perform Westerns with antibodies against ThMFS1 and 3 to determine if they recognize proteins of correct sizes.

Done, we have Western Blots data for each antiserum generated against the selected pair of peptides for each ThMFS1-4 proteins and these data are presented in a new supplement, Figure 2—figure supplement 5, as described above. Consistent with the IFA data the antisera for ThMFS1/3 detect proteins in *T. hominis* infected RK13 cells and spores, whereas the antisera for ThMFS2/4 do not detect any signal in either *T. hominis* infected cells or isolated spores. The apparent Mw of parasite specific bands include sizes (~55 kDa) that matches the theoretical Mw of ThMFS1 and 3, respectively 55.1 kDa and 50.4 kDa.

2) Provide more controls, analyses and information for transport assays.A) Show positive controls for CTP and UTP transport (Figure 3C).

Unfortunately, we are unable to provide these data and respectfully suggest that a new set of experiments is beyond the scope of the present revision. In one previous experiment using a much higher concentration of UTP (50 µM) compared to ATP and GTP (0.5 – 2 µM) we did observe significant transport for UTP by PamNTT3 (the *Parachlamydia* positive control) consistent with published data for this transporter (Haferkamp et al., 2006). We can include these data if it is deemed necessary, but we have no equivalent data for CTP.

B) Provide statistics to show significance for transport of ATP and GTP by MFS proteins (Figure 3 and 4).

Done, we added the information related to statistical tests on the transport data. We performed the same test as done for the NTT transporters (Dean et al., 2018, with significant difference being assessed between *E. coli* cells transformed with empty control vectors and the ThMFS1-4 expressing corresponding vectors. This information was added to the figure legends for Figure 3 and 4. We also performed these tests for the data in Figure 5. We considered as significant differences with p-values < 0.05.

The relevant figure legends (Figure 3, 4 and 5) and the Materials and methods section have been modified accordingly.

C) Test MFS proteins for transport of nucleosides or explain why this is not feasible. The authors had negative results when they tested ThMFS proteins for transport of the nucleoside uridine (Figure 3A), but they could test other nucleosides as well, especially given that the MFS proteins were originally annotated as nucleoside transporters. Additional motivation for testing other nucleosides is that T. hominis appears to have enzymes for converting adenosine to AMP, cytidine to CMP and thymidine to TMP (Heinz et al., 2014). Thus, if the MFS proteins can transport the nucleosides adenosine, cytidine or thymidine, T. hominis might be able to convert these nucleosides into nucleotides.

Unfortunately, and although we agree transport assays with additional nucleosides (or other substrates) would be informative, we are unable to provide these data and respectfully suggest that a comprehensive set of further technically challenging experiments is beyond the scope and timeframe of the present revision. Moreover, it is also already clear from genome analyses that different Microsporidia can contain different repertoires of enzymes needed for nucleoside and nucleotide inter-conversions.

D) Explain how/when ThMFS2 and 4 may be used for purine uptake, given that they appear to have very low expression but are functional in GTP/ATP transport (Figure 3C).

We do not have sufficient data to answer this question with any degree of confidence. As discussed in the manuscript the level of transcription of the *V. culicis* orthologues for ThMFS2 and 4 (Figure 1) are also characterised by the lowest level of transcripts across the 3 tested time points of the infection cycle (Desjardin et al., 2015, and see Figure 2—figure supplement 1).

We have added the following sentence to the main text:

“The functional relevance of the transporters ThMFS2 and ThMFS4 is unclear as we were unable to detect any evidence of their protein expression in our model system".

E) For the time course of ATP and GTP uptake (Figure 4), explain the rationale for using two different *E. coli* strains. *S. rosetta*2(DE3)pLysS should contain native NupG – would this be expected to affect the outcome?

We used two *E. coli* strains and ThMFS ORF with- or without *E. coli* optimised codons, to maximise the expression of the Microsporidia transporters in *E. coli*. Expression of eukaryotic membrane proteins in heterologous bacterial hosts is notoriously difficult but these strains of *E. coli* have a good track record of success. We presented the results (including appropriate controls) with the higher levels of transport. We cited a review paper (Gustafsson et al., 2012) that discussed these issues in the original version of the manuscript.

3) Provide more information about RNAseq expression data.A) Explain how expression data correlates between Figure 2A and 2B- ThMFS4 appears to increase at late timepoints in 2B, but not 2A.

The data in Figure 2A shows the expression of MFS transporters across the lifecycle of the parasite, measured in TPM without normalisation. There is a large difference in the average level of expression for the different MFS transporters. ThMFS4 expression is relatively low on average at 13.6 TPM, compared to ~150 TPM for ThMFS1. ThMFS4 expression does increase in Figure 2A, from a low of 8.7 at T3 to a high of 18.3 at T5, a 2.1x increase. The line for ThMFS4 in Figure 2A tracks this change correctly, but the scale of the plot makes it difficult to observe this trend. This is one of the reasons for including the heatmap in Figure 2B, where expression values (TPM) are normalised based on the average TPM for each gene – the row normalized Z-score. The expression of ThMFS4 at T5 and T6 is higher than the average of 13.6 for that gene, at 18.4 and 18.3 respectively and the heatmap reflects this. All TPM (and corresponding FPKM) values and z-scores (in a new worksheet) are listed in the Figure 2—source data 2.

The legend for Figure 2B has been edited to clarify the z-scores:

“(Z-score – normalised values based on the average TPM for each gene)”.

B) Mention in text that NTT4 has much higher RNAseq expression than the MFS proteins.

We have added a specific statement in the main text in response to this request, which reads as follows:

“Notably ThNTT4 is characterised by the highest level of transcripts among all ThMFS and ThNTT transporters, with mean TPM values of >1000 across the six different time points in the synchronised infections (Figure 2—source data 2)(Dean et al., 2018).”

4) Clarify phylogenetic analysisA) Figure 1—source data 1 is confusing and doesn't seem to match up with numbers in the text (63 members). The "HMMER_Pfam_HFS" tab and "Sheet 1" tab have the same genes listed multiple times. The "Locus tags profile hits" tab has what looks like a non-redundant list of genes, but has a question highlighted in yellow there – perhaps left by the authors who forgot to proof and remove it. The "Grand Total" says 116. What does this refer to?

We thank the reviewer for the careful reading of Figure 1—source data 1).The table lists all Pfam hits for each protein. Some entries have more than one hit – contrast the column “B” listing locus tags with the columns “H” and “I” listing the Pfam domain accession number and domain name, respectively – so the number of lines in the table reflects this, but the table does list 63 distinct proteins/locus tags. The entry SLOPH_1068 was not listed in the main table as this entry had no significant Pfam hit. We have now added this information for completeness in a note at the bottom of the Figure 1—source data 1.

The reviewer is correct – the two extra sheets were not meant to be part of the final submitted version. We apologise for our oversight and have deleted them.

B) Consider Amphiamblys as a Microsporidian.

We prefer to restrict the name Microsporidia to the lineage referred to as the core Microsporidia by Galindo et al., 2018, and also called the canonical or long-branching Microsporidia (e.g. by Bass et al., 2018). We think that the taxonomy of Microsporidia is likely to undergo significant revision in coming years and is beyond the scope of the current manuscript.